

# Inhibition of calcineurin by FK506 stimulates germinal vesicle breakdown of mouse oocytes in hypoxanthine-supplemented medium

Li Wang[1], Yan-Hong Zhen[1,2], Xiao-Ming Liu[1,3], Jing Cao[1], Yan-Ling Wang[1] and Li-Jun Huo[1]

[1] Key Laboratory of Agricultural Animal Genetics, Breeding and Reproduction, Education Ministry of China, College of Animal Science and Technology, HuaZhong Agriculture University, Wu Han, Hu Bei Province, People's Republic of China

[2] Department of Animal Husbandry and Veterinary, Wuhan Agricultural School, Wuhan, Hu Bei Province, People's Republic of China

[3] Reproductive Medicine Center, Second Affiliated Hospital of Wenzhou Medical College, Wen Zhou, People's Republic of China

Corresponding author
Li-Jun Huo, lijunhuo@yahoo.com

## ABSTRACT

Calcineurin (CN) is a serine/threonine phosphatase which plays important roles in meiosis maturation in invertebrate oocytes; however, the role of CN in mouse oocytes is relatively unexplored. In this study, we examined the expression, localization and functional roles of CN in mouse oocytes and granulosa cells. The RT-PCR results showed that the $\beta$ isoform of calcineurin A subunit (Cn A) expressed significantly higher than $\alpha$ and $\gamma$ isoforms, and the expression of Cn A$\beta$ mRNA obviously decreased in oocytes in which germinal vesicle breakdown (GVBD) occurred, while only B1 of calcineurin B subunit (Cn B) was detected in oocytes and stably expressed during oocytes maturation. The following fluorescence experiment showed that Cn A was mainly located in the nucleus of germinal vesicle (GV) stage oocytes and gruanlosa cells, and subsequently dispersed into the entire cytoplasm after GVBD. The decline of Cn A in oocytes suggested that it may play an important role in GVBD. To further clarify the role of calcineurin during meiotic maturation, FK506 (a calcineurin inhibitor) was used in the culture medium contained hypoxanthine (HX) which could keep mouse oocytes staying at GV stage. As expected, FK506 could induce a significant elevation of GVBD rate and increase the MPF level of denuded oocytes (DOs). Furthermore, FK506 could also play an induction role of GVBD of oocytes in COCs and follicles, and the process could be counteracted by MAPK kinase inhibitor (U0126). Above all, the results implied that calcineurin might play a crucial role in development of mouse oocytes and MPF and MAPK pathways are involved in this process.

## INTRODUCTION

Mammalian oocytes arrest at prophase of meiosis I with low MPF (maturation promoting factor) activity in the ovarian follicle. Meiotic resumption from diplotene arrest is

morphologically characterized by germinal vesicle breakdown (GVBD) and is mediated by activation of MPF, which consists of CDK1 and cyclinB. It is well known that the activity of phosphatase oppose to CDK1/cyclin B would affect the net MPF activity (*Adhikari & Liu, 2014*; *Hara et al., 2012*). Cultured with protein phosphatase inhibitor, okadaic acid could induce mouse oocytes to overcome the GVBD inhibition (*Rime & Ozon, 1990*), which also occurs in mouse oocytes microinjected with okadaic or anti-PP1 antibody (*Swain et al., 2003*). However, further study is still needed to identify whether other PP types are involved during prophase I arrest of mouse oocytes.

Calcineurin (CN), protein phosphatase 2B, is a serine/threonine protein phosphatase, which is directly regulated by $Ca^{2+}$/calmodulin in the cell signaling process and consists of catalytic subunit A and regulatory subunit B. Calcineurin is involved in a variety of important activities, such as movements, fertility, egg laying, and growth in *Caenorhabditis elegans* (*Bandyopadhyay et al., 2002*). Calcineurin is also essential for exiting from meiotic arrest in metaphases I and II in *Xenopus* and *Drosophila* oocytes. It releases the *Xenopus* egg meiotic metaphase II arrest by Fizzy/Cdc20, a key regulator of the anaphase-promoting factor6 and Mos-MAP (mitogen-activated protein) kinase pathway (*Mochida & Hunt, 2007*). Moreover, the blockage of calcineurin interrupts the progression of oocyte meiosis at the stage of anaphase I (*Takeo, Hawley & Aigaki, 2010*). Calcineurin along with PP2A is required for normal egg activation of ascidians during fertilization (*Levasseur et al., 2013*). Since ascidians and vertebrates are chordates, it may be inferred that calcineurin may also play an important role in mammalian oocyte maturation and activation.

The mammalian calcineurin subunit A has three isoforms—$\alpha$, $\beta$, $\gamma$, and subunit B is known to have two isoforms and they may have different functions. Knockdown of PPP3CA (calcineurin A$\alpha$) protein expression enhanced vascular endothelial growth factor (VEGF)-stimulated, but not fibroblast growth factor 2 (FGF2)-stimulated and attenuated FGF2-induced MAPK3/1 and AKT1 phosphorylation (*Hand et al., 2005*). When PPP3CB (calcineurin A$\beta$) coexpressed with cAMP-specific 3′, 5′-cyclic phosphodiesterase (PED4D), the degradation of PED4D will be delayed (*Suk et al., 2013*). PPP3CC (calcineurin A$\gamma$) gene might be a susceptibility gene for human bipolar affective disorder (*Mathieu et al., 2008*). PPP3R1 (calcineurin B1) are reported to funciton in alzheimer's disease (*Peterson et al., 2011*) and muscle volume response (*Hand et al., 2005*). While PPP3R2 (calcineurin B2) is known specific expressed in testes (*Ueki, Muramatsu & Kincaid, 1992*).

Calcineurin has been found in many types of mammal somatic cells. However, there is very little information about its expression and distribution in mammal oocytes, especially mouse oocyte. Recently, it is reported that both calcineurin subunit A and B are localized in porcine oocytes during growth and meiotic maturation (*Tumova et al., 2013*). Two potent specific inhibitors of calcineurin has been found which could influence the meiosis of pig oocytes as well as enhance the maturation of growing pig oocytes with partial meiotic competence and delay the maturation of fully grown pig oocytes with full meiotic competence (*Petr et al., 2013*). While, other study shows that calcineurin catalytic subunit is undetectable in M II mouse oocytes and calcineurin inhibitors (CSA, FK506 or CN-inhibitory peptide) cannot prevent meiotic exit (*Suzuki et al., 2010*). This implies that calcineurin plays

no part in mouse oocyte activation, but little is known about expression and function of calcineurin catalytic subunit in mouse oocyte before M II stage. Furthermore, interactions between oocyte and its follicular component *in vivo* have not been considered. Consequently, it cannot be extended to CN's function on mouse oocytes maturation *in vivo*.

As mouse oocytes could undergo spontaneous maturation *in vitro* culture, hypoxanthine (HX) is widely used to keep oocytes staying at GV stage *in vitro* to imitate the *in vivo* oocytes maturation arrest (*Wigglesworth et al., 2013*). Calcineurin inhibitors, tacrolimus (FK506) and cyclosporin A (CSA), are widely used in renal transplantation as immunosuppressive drugs (*Vanhove, Annaert & Kuypers, 2016*) and delay the destruction of cyclins, the global dephosphorylation of M-phase-specific phosphoproteins and the reformation of a fully functional nuclear envelope in study of calcineurin function in oocytes (*Mochida & Hunt, 2007*; *Nishiyama et al., 2007*). Therefore, the aim of this study is to investigate the role of calcineurin in regulation of mouse oocyte meiosis maturation by using FK506 and detect the expression pattern and distribution of calcineurin during oocyte maturation. We found that calcineurin A expressed in mouse oocyte and granulosa cells and inhibition of calcineurin could induce GVBD of oocytes. The results of this study were helpful to understand the regulatory functions of calcineurin on mouse oocytes and provide important information for the regulatory mechanism of folliculogenesis.

# MATERIALS AND METHODS

## Ethics statement and animal feeding regimens

The animal experimental procedures were approved by HuaZhong Agricultural University Institutional Animal Care and Use Committee, Wuhan, China (Approval ID: SCXK (Hubei) 2008_0005). Kunming mice (3–4 weeks old) were obtained from the Centre of Laboratory Animals of Hubei Province (Wuhan, PR China). Mice were housed under controlled temperature (25 °C) and lighting (12 h light/12 h darkness) with food and water ad libitum and sacrificed with cervical dislocation method.

## Isolation and culture of DOs, COCs and follicles

Ovaries were collected from 3 to 4 weeks KM mice, and were minced properly. Later culture medium was added to be convenient for collecting oocytes by straws. A total of 50 fully grown and immature oocytes arrested at the GV stage were cultured in 500 µl M199 medium (Life Technologies Corporation, Rockville, MD) with 4 mM HX or not.

In the same way, ovaries were collected from 3 to 4 weeks KM mice injected with 10 IU PMSG (Sansheng Pharmaceutical Corporation, China) for 44 h, and were punctured by the needles of 1 ml injector. Then COCs with at least two or three layers of granulosa cells or antral follicles (FOs) with a diameter about 300–400 µm were collected from the medium by straws. 50 COCs or FOs were cultured in 500 µl M199 medium with 4 mM HX.

## RT-PCR analysis

RNA from oocytes and granulosa cells was isolated using Rneasy Plus Micro Kit (QIAGEN, Germany) and RNAprep pure cell kit (Tiangen, Beijing China), respectively. Total RNA was transcribed to cDNA with first strand cDNA synthesis kit (Therm, USA). And quantitative

**Table 1  Sequence of primer pairs for quantitative real-time PCR.**

| | | | |
|---|---|---|---|
| Calcineurin A | *Ppp3ca* (isoform α) | Sense primer | 5′-CAACACTCGCTACCTCTTC-3′ |
| | | Antisense primer | 5′-CCATACAGGCGTCATAAA-3′ |
| | *Ppp3cb* (isoform β) | Sense primer | 5′-TAGTGGAGTGTTGGCTGG-3′ |
| | | Antisense primer | 5′-AGTGGTATGTGCGGTGTT-3′ |
| | *Ppp3cc* (isoform γ) | Sense primer | 5′-CCTCTTGCTGCCCTCTTA-3′ |
| | | Antisense primer | 5′-CTTCTCGCTGCCGTAGTC-3′ |
| Calcineurin B | *Ppp3r1* (isoform 1) | Sense primer | 5′-AGGCGAGTTACCCTTTGG-3′ |
| | | Antisense primer | 5′-CTCCGTTGCCGTCTGTG-3′ |
| | *Ppp3r2* (isoform 2) | Sense primer | 5′-CTCCCAGTTCAGCGTCAA-3′ |
| | | Antisense primer | 5′-ATCGCCATCCTTATCCAG-3′ |
| *Actb* | | Sense primer | 5′- CCCATCTACGAGGGCTAT-3′ |
| | | Antisense primer | 5′-TGTCACGCACGATTTCC-3′ |

real-time PCR was carried out using SYBR Green (QuantiFast SYBR Green PCR kit (QIAGEN, Germany)) on a Bio-Rad LC480 real-time PCR system. All primer pairs used for real-time PCR are summarized in Table 1. The amplification efficiencies of primers were obtained by making standard curves and efficiencies of PCR amplification were between 90 and 105% and the variation was less than 5%. The relative quantification of mRNA expression for each calcineurin isoform was estimated using formula $2^{-DDCt}$, according to the comparative Ct method, normalized to the housekeeping gene (*Actb*) in each sample.

## Immunohistochemistry

Mouse ovaries were collected and fixed in Borne's solution for 12–16 h at RT. The fixed mouse ovaries were dehydrated in graded ethanol, dealcoholized with xylene, soaked in methyl salicylate for 12–16 h, and embedded in paraffin. The paraffin-embedded tissues were sectioned into 4-µm thick slices, mounted onto poly-L-lysine-coated slides and dried at 42 °C for 2–4 h in air, subsequently transferred into 60 °C overnight in incubator. The sections were deparaffinized and rehydrated in graded ethanol, soaked in 3% hydrogen peroxide for 30 min to remove endogenous peroxidase and washed in ddH2O two times for 5 min each followed by PBS two times for 5 min each. The sections were boiled in 0.01 M sodium citrate (pH 6.0) three times for 2 min each at 5-min intervals for antigen retrieval. Then the samples were blocked by donkey serum for 30 min at RT, then incubated overnight at 4 °C with anti-Cacineurin A antibody (SC9070; Santa Cruz Biotechnology, USA) (1:100 dilution) or rabbit IgG (Santa Cruz Biotechnology, USA) (1:100 dilution) as control in blocking solution. The sections washed three times for 5 min each in 0.1% Tween-20 in PBS and the samples were incubated with goat anti-rabbit biotin-SP-conjugated antibody (Guge Biotechnology Corporation, Wuhan, China) for 30 min at RT. The sections washed three times for 5 min each in 0.1% Tween-20 in PBS and the samples were incubated with SABC for 30 min at 37 °C. After washing 4–5 times for 5 min each in PBS, the immune reactive signals were detected using DAB Map Kit (Guge Biotechnology Corporation, Wuhan, China) for 3–5 min. The sections washed four times for 5 min each in ddH2O. The samples were stained by hematoxylin for 2 min, differentiated by 5% hydrochloric acid for 8 s, washed by ddH2O three times for 5 min each and washed by PBS three times

for 5 min each to make the stain of nucleus return to blue. The sections were dehydrated in graded ethanol, dealcoholized with xylene, and sealed with neutral resin and cover glass. The sections were put in 37 °C for 24 h–48 h and were observed by microscope with Nomarski optics and digital camera.

## Immunofluorescence and confocal microscopic techniques

Oocytes cultured for 0, 2 or 12 h, at which point most of the oocytes had reached the GV, GVBD or M II stages, were collected for immunofluorescent staining. Oocytes were fixed in 4% paraformaldehyde in PBS (Phosphate Buffered Saline) for 1 h at RT (room temperature). After washing three times in wash buffer (PBS containing 2% BSA), oocytes were treated with 0.5% Triton X-100 in PBS for 1–1.5 h at RT, blocked in PBS containing 5% BSA for 1 h at RT and incubated overnight at 4 °C  with anti-Cacineurin A antibody (Santa Cruz Biotechnology, USA, SC9070) (1:100 dilution) or rabbit IgG (Santa Cruz Biotechnology, USA) (1:100 dilution) as control in blocking solution. Then, oocytes were incubated with a FITC-conjugated goat anti-rabbit antibody (Boster Co., Wuhan, China) (1:100 dilution) in dark after washing three times in 15 min, nuclei were stained in washing buffer containing 10 μg/ml PI (Propidium Iodide) for 10 min. After immunofluorescent staining, oocytes were mounted on slides in DABCO and fluorescent signals were detected using confocal microscope.

Granulosa cells adherent to coverslip were fixed in 4% paraformaldehyde in PBS for 30 min at RT and washed for three times. After treating with 0.5% Triton X-100 in PBS for 15–20 min at RT, granulosa cells were blocked in 5% BSA with PBS for 30 min at RT and incubated overnight at 4 °C with anti-Calcienurin A antibody (Santa Cruz Biotechnology, USA, SC9070) (1:100 dilution) or rabbit IgG (1:100 dilution) as control in blocking solution, then incubated with a FITC-conjugated goat anti-rabbit antibody (1:100 dilution) in dark and washed 3 times in 15 min, later nuclei were stained in washing buffer containing 10 μg/ml PI for 10 min. After immunofluorescent staining, coverslips were mounted on slides with DABCO and fluorescent signals were detected using confocal microscope.

## Western blot analysis

A total of 200 fully grown GV, GVBD and MII oocytes were collected and lysed in 2X SDS sample buffer and boiled in water for 5 min. Total protein was concentrated by 5% SDS-PAGE (SDS-polyacrylamide gel electrophoresis) for 30 min at 90 V, and then was separated by 12% SDS-PAGE for 90 min at 120 V and electrophoretically transferred to a PVDF membrane for 1 h at 200 mA. After washing in TBS (Tris–HCl Buffer Saline) for 3 times, the membrane was blocked in TBST (TBS with 0.1% Tween 20) containing 5% skim milk for 1–2 h, and then incubated with a rabbit anti-Calcineurin A antibody (Santa Cruz Biotechnology, USA, SC9070) (1:200 dilution) or a mouse anti-$\beta$-actin (Santa Cruz Biotechnology, USA, SC69879) (1:1000 dilution) in TBST overnight at 4 °C. After washing 3 times in TBST, the membrane was incubated with an HRP-conjugated anti-rabbit or mouse secondary antibody (Boster Corporation, Wuhan, China) (1:2000 dilution) in TBST for 1 h at RT. The membrane was washed 3 times with TBST and processed with the ECL (Enhanced Chemilumidescence) detection system. ACTB was detected as an internal control.
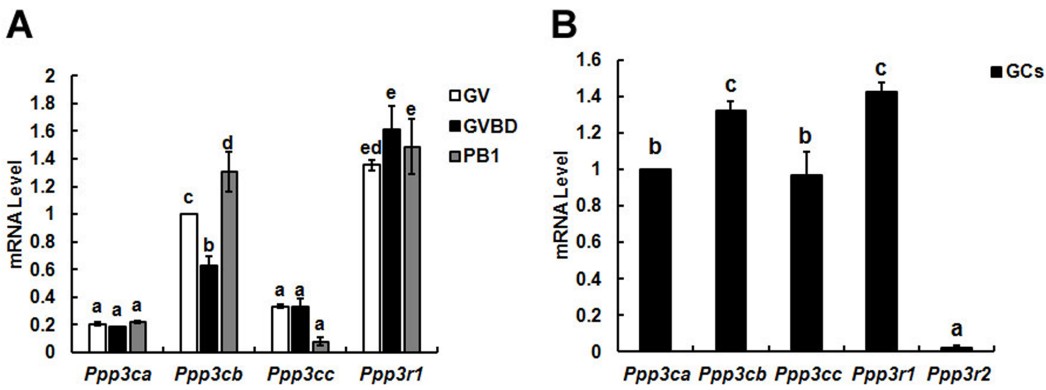

**Figure 1** Relative mRNA expression of calcineurin isoforms in mouse oocytes (A) and granulosa cells (B). The relative mRNA levels represent the amount of mRNA expression normalised to Actb. The comparison of calcineurin mRNA expression was made between each oocytes' stage and oocytes in one stage, so was in granulosa cells. The statistical difference between two different lowercase letters represents $P < 0.05$, such as "a" and "b," "a" and "c," or "b" and "c" represents $P < 0.05$.

## MPF activity assay

After treating with or without FK506 for 12 h, 100 oocytes were lysed with 30 µl of radio immune precipitation assay (RIPA) buffer supplemented with a protease inhibitor cocktail (Sigma). Samples were lysed on ice for 5–10 min, and then centrifuged at 4 °C at 12,000 rpm for 15 min. The supernatant was collected and stored at −80 °C until use. The level of MPF was determined using the Mouse MPF ELISA Kit (Kexing Biotechnology Company, China) following the manufacturer's protocol and each sample was tested with three repeats. All other chemicals were purchased from Sigma Aldrich, unless stated otherwise.

## Statistical analysis

Each experiment was repeated at least three times and data were presented as mean ± SEM (standard error of the mean). Differences between groups were analyzed using one-way ANOVA followed by Tukey's Honest Significant Difference (HSD) test using SPSS (Version 17.0; SPSS, Chicago, IL). $P$ values < 0.05 were considered to be statistically significant.

## RESULTS

### Identification of the mRNA expression level for isoforms of calcineurin subunits

To identify the different expression pattern of calcineurin' subunits during oocytes meiotic maturation, three developmental stages (GV, GVBD and PB1 extrusion) oocytes and granulosa cells were isolated for detection.

Three isoforms ($\alpha$, $\beta$ and $\gamma$) of calcineurin A subunit mRNA were detected in oocytes and the mRNA level of $\beta$ isoform was significantly higher than $\alpha$ and $\gamma$ during meiotic maturation. While only B1 isoform of calcineurin B was found in oocytes and expressed during oocyte maturation constantly (Fig. 1A). Similar with oocytes, three isoforms of calcineurin A subunit were detected in granulosa cells and the mRNA level of Cn A$\beta$ was significantly higher than Cn A$\alpha$ and Cn A$\gamma$. However, mRNAs of two isoforms of calcineurin B (B1 and

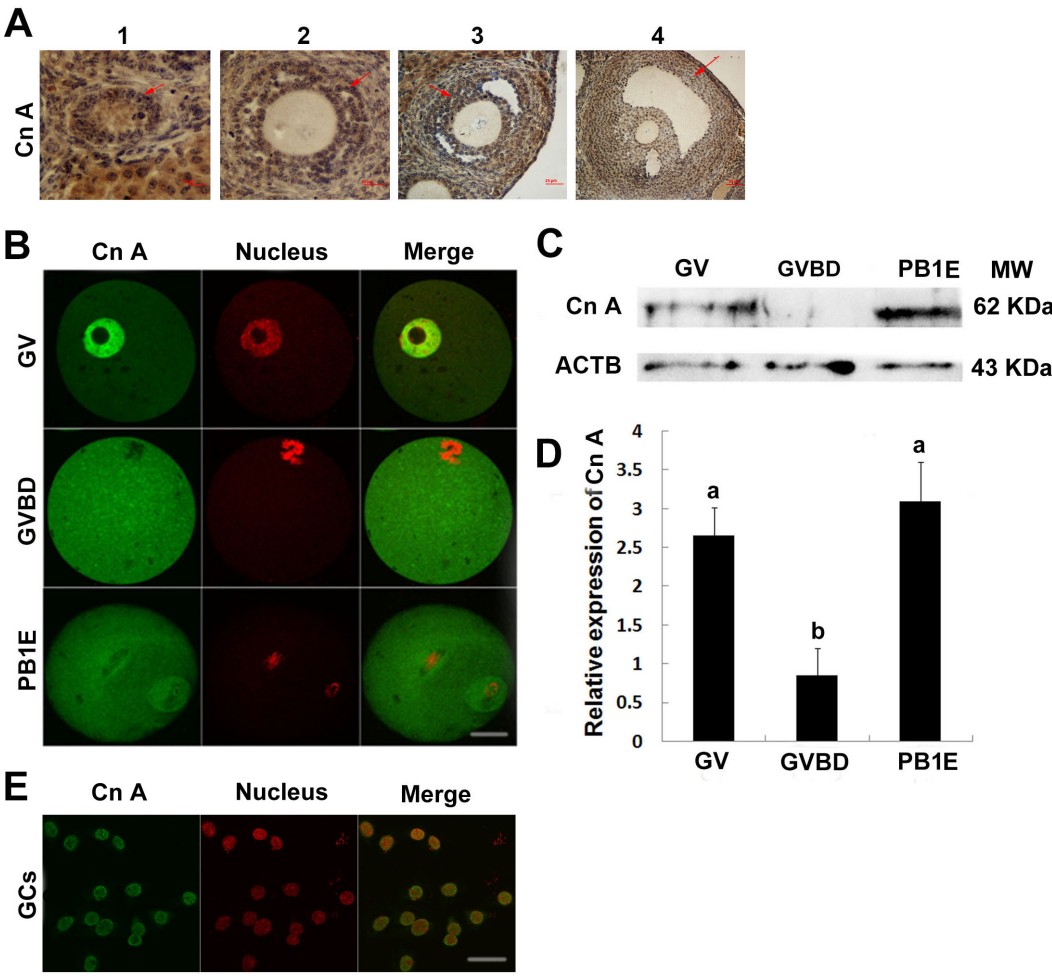

**Figure 2  Expression and location of calcineurin A subunit in follicles, oocytes and granulosa cells.**
(A) Calcineurin A subunit was detected by immunohistochemistry in different stage of follicles (1, primordial follicle; 2, primary follicle; 3, small antral follicle; 4, big antral follicle), which were pointed out by red arrows. (B) Cellular localization of calcineurin A subunit was detected by immunofluorescence in GV, GVBD and PB1 extrusion oocytes. (C, D) Calcineurin A subunit protein expression was confirmed by Western blot in GV, GVBD and PB1 extrusion oocytes and the protein ratios were analyzed according to the western blot results. "a" and "b" represents $P < 0.05$. (E) Cellular localization of calcineurin A subunit was detected by immunofluorescence in granulosa cells. Scale bar, 20 μm.

B2) subunit were both detected in granulosa cells and the mRNA level of B1 was significantly higher than B2 (Fig. 1B).

## Localization and protein expression of calcineurin A in oocytes and granulosa cells

The above study showed that Cn A$\beta$ mRNA level decreased significantly after GVBD, which suggested Cn A might play important roles during GVBD. Thus, we examined the localization and expression of Cn A in mouse ovaries. Positive reaction of calcineurin A was found in different stages of follicles (Fig. 2A). Furthermore, immunocytochemical evaluation showed the present of calcineurin A subunit in different stages of oocytes and granulosa cells and

was expressed in the nucleus of fully grown GV stage oocytes and granulosa cells, while homogenously dispersed into cytoplasm after GVBD (Figs. 2B and 2E). Western blot analysis showed that the expression of calcineurin A subunit at GV and PB1 extrusion stages of oocytes was significantly higher than GVBD stage oocytes (Figs. 2C and 2D), which was consistent with the mRNA expression of calcineurin A.

## Inhibition of calcineurin by FK506 could induce the GVBD of oocytes

To further study the functional role of calcineurin in the development of oocyte, DOs (denuded oocytes) were incubated in basic medium (without HX) containing different concentration of FK506. Since 50 µM FK506 could lead to abnormal cell morphology of oocytes (data not showed), we used 5 µM and 25 µM FK506 to culture DOs and examined the GVBD rate after 2 h. The results showed that 5 µM and 25 µM could not disturb the GVBD of oocytes (Fig. 3A) while, when DOs were cultured in medium containing 4 mM HX for 12 h, 25 µM FK506 could significantly increase the GVBD% ($54.91 \pm 1.47\%$) compared to the control group ($37.93 \pm 2.31\%$), as well as 5 µM FK506 ($45.81 \pm 1.59\%$) (Fig. 3B).

As calcineurin was expressed both in the oocytes and granulosa cells, we supposed that calcineurin may participate in the communication between oocyte and cumulus cells. To confirm the hypothesis, COCs (cumulus oocyte complexes) were cultured in medium with HX and FK506, and GVBD rates were examined after 22 h. COCs cultured with FK506 and HX showed an significantly increase of GVBD% ($83.15 \pm 3.28\%$) compared to control group ($58.13 \pm 6.74\%$), and showed no significant difference with FSH group ($88.85 \pm 6.16\%$) (Fig. 3C). However, inhibition of calcineurin by FK506 had no effect on cumulus expansion (Fig. 3D). These results suggested that calcineurin might play an important role in maintaining the oocytes stayed at GV stage. Antral follicles, which had a diameter about 300–400 µm and contained about the same condition of COCs as we cultured, were cultured by 25 µM FK506 to confirm that 25 µM FK506 also could induce the GVBD of oocytes from the antral follicles after culture for 22 h (Fig. 3E).

## FK506 increased MPF level in oocyte which could be reduced by U0126

It has been well established that MPF and MPAK are important pathways in meiotic maturation. To further test whether there was feedback regulation between MPF and calcineurin in oocytes, MPF concentration was detected by ELISA. The results showed that the MPF concentration increased significantly after 25 µM FK506 treating for 12 h and partially decreased by MAPK inhibitor U0126 (Fig. 4). These results suggest that both MAPK and MPF are associated with the regulation of calcineurin in oocytes.

## DISCUSSION

Previous studies have demonstrated that calcineurin is required for *Xenopus* egg meiotic metaphase II resumption and is necessary for the completion of oocyte meiosis in Drosophila (*Mochida & Hunt, 2007*; *Takeo, Hawley & Aigaki, 2010*). However, little information exists about the role of calcineurin in mammal oocytes maturation. Therefore, we detected the expression, location and function of calcineurin in mouse ovaries. The

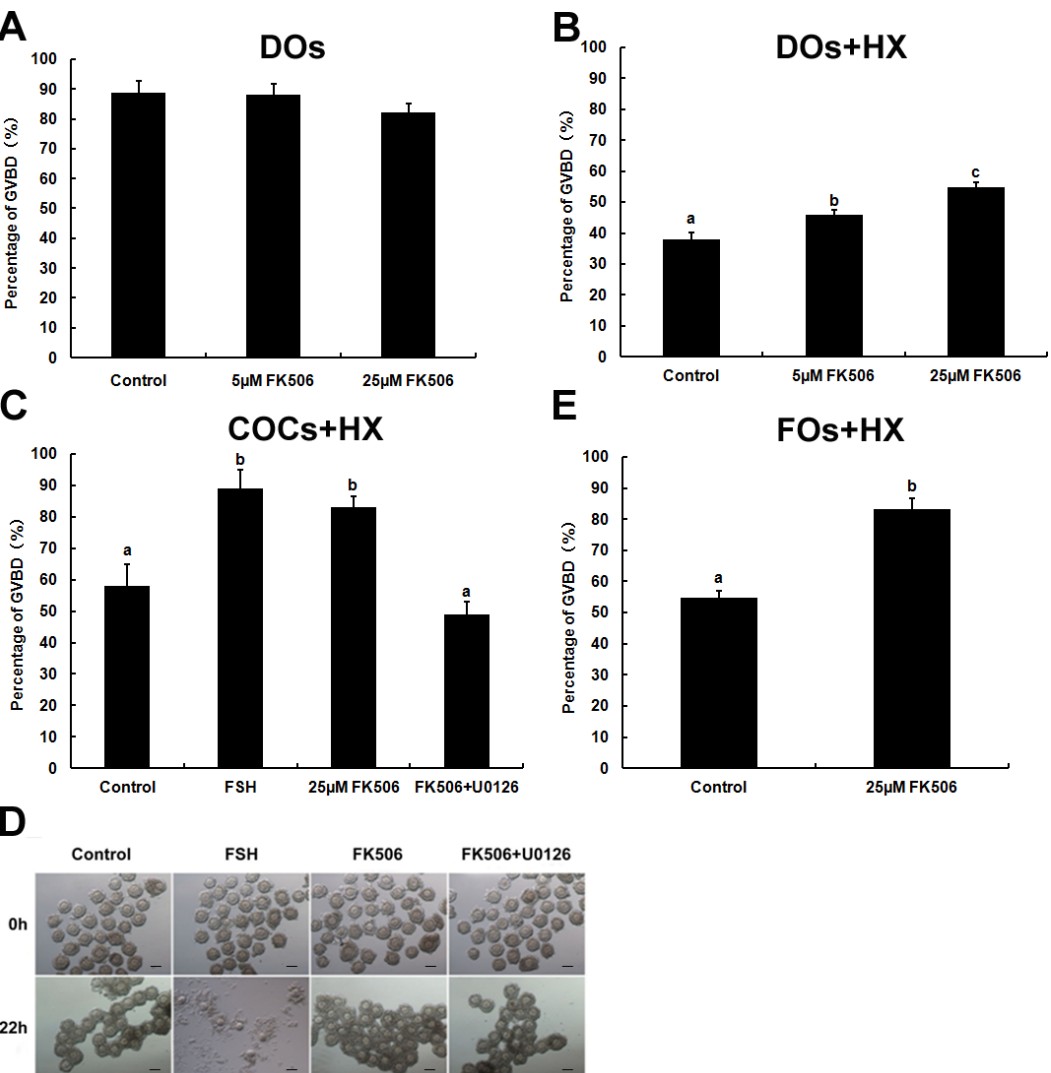

**Figure 3** **The effects of FK506 on GVBD rate of oocytes and cumulus expansion.** (A, B) The percentage of GVBD in oocytes cultured with different concentration of FK506 in medium with or without 4 mM HX. (C, D) The GVBD percentage of oocytes from COCs cultured with FSH, 25 μM FK506 or U0126 + 25 μM FK506 in medium with 4 mM and the cumulus expansion was examined. (E) The GVBD percentage of oocytes from FOs cultured with 25 μM FK506 in medium contain 4 mM. Data presented as mean ± standard error of the mean, from three independent experiments. Different lowercase letters indicated statistical difference ($P < 0.05$).

results showed that calcineurin was expressed in oocytes and granulosa cells of all stages of follicles and inhibition of calcineurin could induce mouse oocytes GVBD, which indicated that calcineurin is an important regulator in mouse ovary.

An important aspect of our study was to detect the expression of calcineurin in mouse ovary. The transcription of genes for various isoforms of calcineurin in mouse oocytes was thoroughly studied. The mRNA for three isoforms of subunit A of calcineurin was detected and expressed in mouse oocytes and the mRNA expression level of β isoform was significantly higher than α and γ isoform, while, only isoform β and γ were detected in pig oocytes

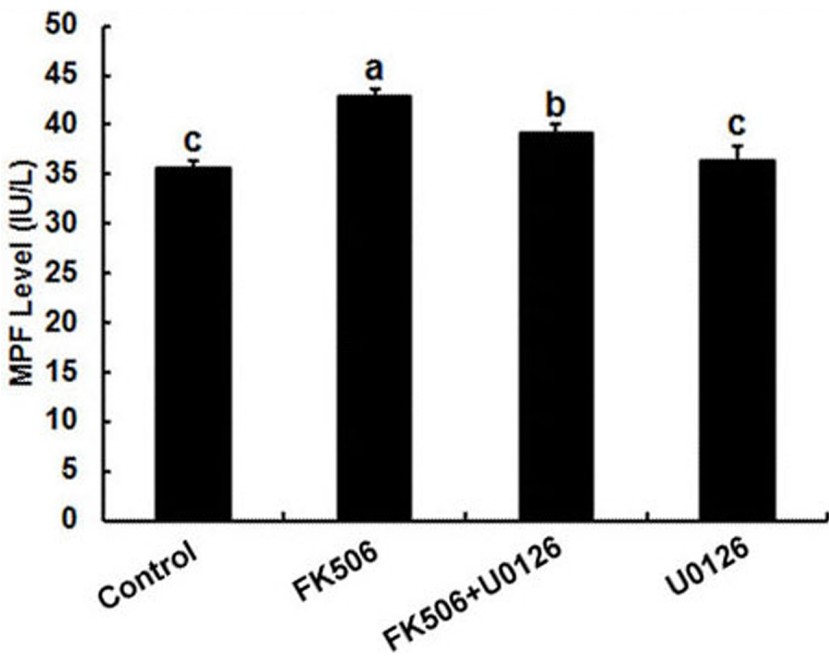

**Figure 4  Inhibition of calcineurin by FK506 increased the concentration of MPF in oocytes.** Concentration of MPF was detected after culturing in 25 µM FK506 for 12 h. Data presented as mean ± SEM, from three independent experiments. "a" and "b," "a" and "b" or "b" and "c" represents $P < 0.05$.

and cumulus cells (*Tumova et al., 2013*). Furthermore, the expression of isoform $\beta$ of calcineurin subunit A decreased significantly after GVBD of mouse oocytes, which was consistent with western blotting analysis that the protein expression of calcineurin A subunit at GV stage oocytes was significantly higher than GVBD stage oocytes. Besides, the isoform 1, but not isoform 2, of calcineurin subunit B was detected in mouse oocytes, as opposed to porcine oocytes where testis-specific calcineurin isoform 2 was detected (*Tumova et al., 2013*). Cn B mRNA was expressed constantly during oocytes maturation. These results suggested that calcineurin might play an important role in GVBD of mouse oocytes, especially the isoform $\beta$ of calcineurin subunit A.

Calcineurin A subunit was localized mainly in the cortex of porcine oocytes during meiotic maturation and started to homogenise during the germinal vesicle breakdown (*Tumova et al., 2013*). This localization might due to its special function on cortical granule excocytosis during pig oocyte activation (*Tůmová et al., 2016*). However, the present results showed Cn A was mainly localized in germinal vesicle of GV stage mouse oocyte and homogenously dispersed into cytoplasm after GVBD. It suggested that calcineurin might function directly in nucleus as an transcriptional regulation factor. Moreover, it also occurred in the nucleus of granulosa cells. We hypothesized that calcineurin might also affect the maturation of oocyte through the granulosa cells and oocyte communication. As expected, the GVBD rate of oocytes from COCs in FK506 group increased significantly compared to control group, so was in oocytes from follicles. But FK506 had no significant effect on inducing the expansion of COCs. It is reported that PDE4D is expressed in granulosa cells but not in oocytes and inhibition of PDE4D induces oocyte maturation by increasing cAMP levels

in granulosa cells (*Tsafriri et al., 1996*). Recent study showed that calcineurin could bind directly to and inhibit the proteosomal degradation of PDE4D, and the levels of PDE4D expression were potentiated when it coexpressed with CnA $\beta$ (*Zhu et al., 2010*). Thence, these findings indicated that inhibition of calcineurin by FK506 might induce oocyte maturation through promoting the degradation of PDE4D in cumulus cells.

It has been well established that MPF and the mitogen activated protein kinase (MAPK) are important pathways in meiotic maturation. Though MPF and MAPK activities are dispensable for breakdown of the oocyte GV membrane (*Endo et al., 2006*), it is not required for the activation of MPF during the first meiosis (*Sugiura et al., 2006*), when injected into GV stage mouse oocytes, the changes in activity of a cytoplasmic MPF are capable of inducing GVBD and PB1 extrusion (*Nakano & Kubo, 2000*). Conditional knockout ERK1/2 in granulosa cells *in vivo* resulted in LH-induced oocyte resumption of meiosis, ovulation, and luteinization failure (*Fan et al., 2009*). The activation of MAPK by FSH in cumulus cells can induce GVBD of mouse oocytes (*Fan et al., 2004*). Protein phosphatases can play an important role in mechanisms of MPF activation and inactivation. Our results showed that FK506 increased GVBD rate and MPF level in mouse DOs which could be reduced by U0126, the specific inhibitor of MAPK. This was consistent with the report that MAPK could override phosphodiesterase inhibitor 3-isobutyl-1-methylxanthine (IBMX) induced mouse oocytes maturation arrest (*Choi et al., 1996*). The functions and relationship between MAPK and MPF are complicate in oocyte development. MAPK and MPF play crucial roles in spindle assembly checkpoint, increased activity of MAPK could cause disturbance of microtubules and irregularly pull chromosomes disperse over the spindle and activate spindle assembly checkpoint. Interestingly, during spontaneous activation, MAPK first decline and then increase and the activity of MPF fluctuates similarly but always changes ahead of the MAPK activity (*Cui et al., 2012*). In addition, activation of MAPK and MPF by c-erbB and c-myb induced oocyte GVBD (*Zheng et al., 2012*; *Zheng et al., 2008*). Inhibition of MAPK by U0126 could decrease the GVBD rate and MPF expression induced by FK506. FK506 could induce a rapid and transient increase of ERK1/2 phosphorylationin rat renal mesangial cells (*Akool El et al., 2012*), we can speculate that activation of MAPK and MPF pathways might involve in the FK506 induced oocyte maturation.

Thus, we speculated that inhibition of calcineurin by FK506 might induce mouse oocyte GVBD through activating MPF and MAPK pathways and inhibiting the PDE4D pathway. Calcineurin may serve as a key regulatory factor in the regulation of development of mouse oocytes *in vivo*. However, further studies will be needed to validate the biological significance of calcineurin in the development of oocytes.

### Funding

This work was supported by the National Natural Science Foundation of China (Grant No. 31171378) and the Fundamental Research Funds for the Central Universities (Program NO. 2014PY045). The funders had no role in study design, data collection and analysis, decision to publish, or preparation of the manuscript.

## Grant Disclosures

The following grant information was disclosed by the authors:
National Natural Science Foundation of China: 31171378.
Fundamental Research Funds for the Central Universities: 2014PY045.

## Competing Interests

The authors declare there are no competing interests.

## Author Contributions

- Li Wang conceived and designed the experiments, performed the experiments, analyzed the data, contributed reagents/materials/analysis tools, wrote the paper, prepared figures and/or tables.
- Yan-Hong Zhen, Xiao-Ming Liu, Jing Cao and Yan-Ling Wang performed the experiments, contributed reagents/materials/analysis tools.
- Li-Jun Huo conceived and designed the experiments, analyzed the data, wrote the paper, prepared figures and/or tables, reviewed drafts of the paper.

## Animal Ethics

The following information was supplied relating to ethical approvals (i.e., approving body and any reference numbers):

The animal experimental procedures were approved by the Huazhong Agricultural University Institutional Animal Care and Use Committee, Wuhan, China (Approval ID: SCXK (Hubei) 2008_0005).

## Data Availability

The raw data has been supplied as a Supplementary File.

## Supplemental Information

Supplemental information for this article can be found online at http://dx.doi.org/10.7717/peerj.3032#supplemental-information.

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
