# Peer review of "Inhibition of calcineurin by FK506 stimulates germinal vesicle breakdown of mouse oocytes in hypoxanthine-supplemented medium"

_PeerJ, doi:10.7717/peerj.3032_

## Round 0.1 · original submission · Major Revisions

From the reviewers' comments you can see, your writing and the interpretation is rough and needs a comprehensive re-check. Please revise and response to the reviewers to improve your manuscript.

Reviewer 1 ·

Basic reporting

No Comments

Experimental design

The manuscript originally investigated the roles of calcineurin in mammalian oocytes. The authors performed rigorous investigation in a high standard to reveal that calcineurin might play a crucial role in meiotic arrest of mouse oocytes by regulating MPF and MAPK pathway. The methods described with sufficient detail to replicate.

Validity of the findings

The data is robust and statistically which well supported the conclusions with originality.

Additional comments

The work by Wang et al tried to investigate the roles of Calcineurin (CN) in mammalian oocytes. The authors found that the β isoform of calcineurin A subunit (Cn A) expressed significantly
higher than α and γ isoforms, and the expression of Cn A β mRNA decreased obviously in oocytes when germinal vesicle breakdown (GVBD), and only B1 of calcineurin B subunit (Cn B) was detected in oocytes and expressed constantly during oocytes maturation. Cn A was mainly located in nucleus at GV stage oocytes and gruanlosa cells, and subsequently dispersed into the entire cytoplasm after GVBD. Further, the authors clarified the role of calcineurin during meiotic maturation by using FK506 (a calcineurin inhibitor). FK506 could induce a significant elevation of GVBD rate and increased the MPF level of denuded oocytes (DOs). Furthermore, FK506 could also play an induction role of GVBD of oocytes in COCs and follicles, and the process could be counteracted by MAPK kinase inhibitor (U0126). The authors concluded that calcineurin might play a crucial role in meiotic arrest of mouse oocytes by regulating MPF and MAPK pathway. The data are convincing, and the manuscript is proper for publication in the journal.

Reviewer 2 ·

Basic reporting

The manuscript is roughly prepared. The authors should examine the whole manuscript since some mistakes were shown in the manuscript like locatedin (Line 18), Drosophilaoocytes (Line 45), γand (Line 53), vitroto (Line 69) and so on.

Experimental design

No Comments

Validity of the findings

No Comments

Additional comments

Comments to the author(s) in this manuscript “Inhibition of calcineurin by FK506 stimulates germinal vesicle breakdown of mouse oocytes in hypoxanthine-supplemented medium”.

Calcineurin (CN) is the serine/threonine phosphatase which regulated by Ca and Calmodulin, and it also involved in invertebrate oocyte maturation. In mammalian oocytes, roles of calcineurin is still largely uncovered. The authors investigated the expression and localization of CN by real-time PCR, immunofluorescent and immunohistochemistry in oocytes and granulosa cells, respectively. Next, FK506 was used in the culture medium contained hypoxanthine to clarify the role of calcineurin during meiotic maturation. Their results show that β isoform of Cn A is significantly higher than α and γ isoforms, and mRNA of Cn A β was decreased obviously in GVBD stage oocytes. Cn A displayed the dynamic localization during oocyte maturation. They also found FK506 increased the rate of GVBD and the MPF abundance in Dos in HX supplement culture model, and this effects were counteracted by use of U0126 in oocytes or COCs. This research may facilitate further investigation in mammalian oocytes maturation. However, the manuscript need to solve the following issues.

1. There is no method of immunohistochemistry in the part of Materials and Methods.
2. In Fig. 1, the letter of “abc, et,al” which indicates the statistic difference should be labeled successively from left to right in a same bar graph.
3. In Fig. 2 B and E, the author should label the objects of staining rather than the name of the stain.
4. In Fig. 3 B, C and E, the letter of “abc” which indicates the statistic difference should label successively from left to right in a same bar graph. Scale bar should be labeled on Fig. 3D
5. In Fig. 4 U0126 but not UO126.
6. How did the authors determine the concentration of FK506, is the inhibitory effect specific?

·

Basic reporting

PP1 (protein phosphatase 1) has been found to be involved during prophase I arrest of mouse oocytes by regulating MPF activity.The current study found another PP type, protein phosphatase 2B(Calcineurin), also take part in the process. The role and regulation of calcineurin has been studied well in vertebrate oocytes, but no further study in mammalian oocytes. The results in the study make a good complement based on solid and convincing data and also made good discussion on the results, so it is a valuable study.
After several times of careful and thorough reading, I think the experimental design is well-conceived,the operation & manipulation is proficient and creditable, and the discussion is pertinent and perceptive. However, it has to be modified and improved before it can be accepted. Specific points are as follows:
1.Figure 1, first part Result section
Since transcription activity ceased during GV to MII in meiosis of mouse oocytes, how would you explain the increase of mRNA level of PPP3CB and PPP3R1 during this period?
As described, lowercase letters indicated statistical difference, but I cannot understand what do they mean and what is the difference between different letters, also in other figures.
2.Figure 2
According to Figure 1, the isform 1 of calcineurin subunit B was detected in mouse oocytes, and the mRNA level seems much higher than isforms of subunit A. Since one important aspect of your study was to detect the expression of calcineurin ,not only subunit A, I think you can further detect the subcellular localization and protein level of subunit B in oocytes.
3.Figure 3.
The main phenotype of mouse oocytes treated with FK506 was the increase of GVBD rate. Did you make the judgment of oocytes at GVBD stage by immunofluresence or just observation using light microscope? Was there any morphological evidence?
4.How about the mRNA or protein level of Calcineurin in oocytes treated with FK506 at different concentration? In another word, please testify the effect on the target of the specific inhibitor.

Experimental design

No Comments

Validity of the findings

No Comments

Additional comments

None

Reviewer 4 ·

Basic reporting

No Comments.

Experimental design

No Comments.

Validity of the findings

No Comments.

Additional comments

In the manuscript entitled “Inhibition of calcineurin by FK506 stimulates germinal vesicle breakdown of mouse oocytes in hypoxanthine-supplemented medium”, Wang et al. investigated the expression, localization and functional roles of calcineurin in mouse oocytes and granulosa cells. Overall, the manuscript is well written (while with some minor grammatical and format errors), and results are clearly demonstrated and technically sound as well. It’s also interesting and significant since very limited studies have been focused on the role of calcineurin in mammalian oocyte maturation. However, some details of the results were not clearly exhibited, and although the finding is novel, insufficient discussion dampens the impact of this study. Detailed concerns and issues based on the current manuscript are outlined below.

1. Abstract, “decreased obviously in oocytes when germinal vesicle breakdown (GVBD)” needs revision.

2. Need spaces: "(Cn A)expressed significantly", "mainly locatedin", "contained hypoxanthine (HX).As expected,FK506could induce", "(U0126).Above all, the", "inhibition(Rime & Ozon 1990)", "antibody(Swain et al. 2003)", "in vitroto imitate", etc.

3. It is suggested to change "calcineurin inhibitors CsA" to "calcineurin inhibitors CSA".

4. "before MⅡstage, Furthermore interactions" needs revision.

5. The study aim "Therefore, the aim of this study is to investigate the role of calcineurin in regulation of mouse oocyte meiosis maturation by using FK506 to culture oocytes alone or with cumulus cell and thento detect the expression pattern and distribution of calcineurin during oocyte maturation." was not clear or easily understandable. It is suggested to rewrite with short and clear sentences.

6. Please be consistent, "HuaZhong Agriculture University" and "Huazhong Agricultural University".

7. Please specify why chose immature mice but not adults in this study? "10 IU PMSG" seems overdosed for these immature mice?

8. Please add manufacture to each reagent used in this study, e.g. PMSG.

9. Need revision: "M199 medium ((Life Technologies Corporation, Rockville, MD))", "kit (Therm, Co. USA)".

10. Please pay attention to nomenclature, including in the figures and tables. mRNA and gene symbol should be in italic. For mouse ones, all should be in lower case except the first letter. β-actin can be updated to Actb.

11. Fig. 1A, "Three isoforms (α,β and γ) of calcineurin A subunit mRNA were detected in oocytes and the mRNA level of β isoform was significantly higher than α and γ during meiotic maturation", how to rule out that different expression levels/qRT-PCR result was due to various primer efficiencies? Since even same transcript and cDNA, different primer pairs could form diverse signal intensities during qPCR.

12. Fig. 2, please add the molecular weights of detected proteins in the presented blots. Please double check "E: Cellular localization of calcineurin A subunit was detected by immunofluorescence in granulosa cells", "Granulosa cells ... incubated overnight at 4°C with anti-Calcienurin A antibody (1:100 dilution), anti-Cacineurin B antibody (Santa Cruz Biotechnology, USA, SC33166)", and specify it was CnA and/or CnB examined in granulosa cells?

13. Fig.3, "medium with 4 mM and" and "in medium contain 4 mM. Data" need revision to make the sentence intact. Please add scale bar(s) to Fig. 3D.

14. In this study, authors discovered a novel finding that MPF activity was increased significantly after FK506 treatment, while decreased by MAPK inhibitor U0126. However, the functions and relationship between MAPK and MPF, two most well studied cell cycle regulators in oocytes, was not sufficiently discussed. It seems more appropriate to discuss more, e.g. in spindle and microtubule (PMID: 20948319, PMID: 22384134), in activation and developmental potency (PMID: 21554769), etc.

15. It is understandable to make the discussion that "we can speculate that MAPK and MPF pathway might involve in the FK506 induced oocytes maturation", however, it seems not very appropriate to draw the conclusion in the abstract that "calcineurin might play a crucial role in meiotic arrest of mouse oocytes by regulating MPF and MAPK pathway" since no data in the present study showed that calcineurin is the upstream of MPF and MAPK ("by regulating MPF and MAPK pathway").

Reviewer 5 ·

Basic reporting

Research in this paper presents data on the role of Calcineurin (CN) in maturation focused on meiosis of mouse oocytes and the role of granulosa cells in this process. The authors examined the expression, localization and functional roles of CN in mouse oocytes and granulosa cells and showed that β isoform of calcineurin A subunit (Cn A) expressed significantly higher than α and γ isoforms and that the expression of Cn Aβ mRNA decreased in oocytes during germinal vesicle breakdown (GVBD).

Experimental design

Standard methods are used to generate the data including RT-PCR analysis, immunofluorescence and confocal microscopy techniques, western blotting analysis, and an MPF activity assay. While these techniques generally produce good results, the paper lacks the necessary control experiments which especially is important for the immunofluorescence analysis, as a large part of the data rely on immunofluorescence. More control experiments are needed to eliminate potentially false positives and false negatives to validate the results. This is especially important since the authors report differences for porcine and mouse oocytes.

Validity of the findings

General points:
In the text the authors refer to “mammalian oocytes” but it would be better to refer to mouse oocytes, as there are significant differences between different species and generalizations should be avoided.

The text is very difficult for the reader to understand because of awkward wording and sentence constructions. In addition, the text contains an unusual amount of mis-spelled words. The text is filled with spelling mistakes. This is true for all sections of the paper. The text needs significant revisions in all sections of the paper. Words are used wrongly such as “constantly” and others.

Abstract:
The abstract is difficult for the reader to follow and should be structured more logically to concisely pose the questions to be addressed, the relevance, and the rationale for the experiments, followed by results. For example, the reader might not understand why hypoxanthine is used in the culture medium and why calcineurin might play a role in regulating MPF and MAPK. More clarity is needed.

Introduction:
As indicated above, “mammalian oocytes” should be explained better, as the mouse in many aspects does not represent all mammalian species and generalizations should be avoided.

The introduction should also focus more clearly on the function(s) of calcineurin and explain the differences between the different isoforms.

The text should comply with commonly used scientific language and avoid casual language such as “breaks” the Xenopus…. (line 45), “ascidians are a sister group”…. (line 50) and numerous others. Using more scientific language is an absolute necessity. Some sentences do not make sense such as the sentences in lines 63 to 67.

Lines 76 to 80: The authors do not convincingly address the role of calcineurin in regulation of processes in oocytes. More specifics and more details are needed.

Materials and methods:
As indicated above, commonly used scientific language should be applied. For example, “chopped” (line 90) is not adequate. Other terms are not used properly.

More specifics should be provided for the supply sources (such as “Sigma” in line 150 and others).

Results:
Inconsistencies should be corrected such as “PB1” (line 163) which is not considered a developmental stage. The text should be corrected for accuracies.

Statements such as “be more important” (line 180) are unfounded and should be modified (the levels of intensity do not reflect importance).

The conclusion in lines 230-231 should also be modified to more specifically explain the suggestion that both MAPK and MPF are associated with the regulation of calcineurin in oocytes.

Discussion:
The sentence in lines 241-243 should be modified to provide more specifics and details on “the results showed that calcineurin was expressed in all stages of follicles and involved in oocytes GVBD process, which indicated that calcineurin is an important regulator in mouse ovary”.

The differences between porcine oocytes and mouse oocytes regarding calcineurin A should be discussed in more detail, not just by stating them but providing more insights on the possible reasons and underlying mechanisms in both species.

The entire text of the discussion should be adjusted for more scientific language to eliminate phrases such as “it is said” (line 265).

The aspects of MAPK and MPF should be explained in more detail and it is not clear how MAPK and MPF “pathways” play a role.

Overall, the paper presents new data but significant changes are needed as outlined above.

Additional comments

General points:
In the text the authors refer to “mammalian oocytes” but it would be better to refer to mouse oocytes, as there are significant differences between different species and generalizations should be avoided.

The text is very difficult for the reader to understand because of awkward wording and sentence constructions. In addition, the text contains an unusual amount of mis-spelled words. The text is filled with spelling mistakes. This is true for all sections of the paper. The text needs significant revisions in all sections of the paper. Words are used wrongly such as “constantly” and others.

Abstract:
The abstract is difficult for the reader to follow and should be structured more logically to concisely pose the questions to be addressed, the relevance, and the rationale for the experiments, followed by results. For example, the reader might not understand why hypoxanthine is used in the culture medium and why calcineurin might play a role in regulating MPF and MAPK. More clarity is needed.

Introduction:
As indicated above, “mammalian oocytes” should be explained better, as the mouse in many aspects does not represent all mammalian species and generalizations should be avoided.

The introduction should also focus more clearly on the function(s) of calcineurin and explain the differences between the different isoforms.

The text should comply with commonly used scientific language and avoid casual language such as “breaks” the Xenopus…. (line 45), “ascidians are a sister group”…. (line 50) and numerous others. Using more scientific language is an absolute necessity. Some sentences do not make sense such as the sentences in lines 63 to 67.

Lines 76 to 80: The authors do not convincingly address the role of calcineurin in regulation of processes in oocytes. More specifics and more details are needed.

Materials and methods:
As indicated above, commonly used scientific language should be applied. For example, “chopped” (line 90) is not adequate. Other terms are not used properly.

More specifics should be provided for the supply sources (such as “Sigma” in line 150 and others).

Results:
Inconsistencies should be corrected such as “PB1” (line 163) which is not considered a developmental stage. The text should be corrected for accuracies.

Statements such as “be more important” (line 180) are unfounded and should be modified (the levels of intensity do not reflect importance).

The conclusion in lines 230-231 should also be modified to more specifically explain the suggestion that both MAPK and MPF are associated with the regulation of calcineurin in oocytes.

Discussion:
The sentence in lines 241-243 should be modified to provide more specifics and details on “the results showed that calcineurin was expressed in all stages of follicles and involved in oocytes GVBD process, which indicated that calcineurin is an important regulator in mouse ovary”.

The differences between porcine oocytes and mouse oocytes regarding calcineurin A should be discussed in more detail, not just by stating them but providing more insights on the possible reasons and underlying mechanisms in both species.

The entire text of the discussion should be adjusted for more scientific language to eliminate phrases such as “it is said” (line 265).

The aspects of MAPK and MPF should be explained in more detail and it is not clear how MAPK and MPF “pathways” play a role.

Overall, the paper presents new data but significant changes are needed as outlined above.

---

## Round 0.2 · accepted · Accept

All the issues were addressed and the reviewers suggest publication.

·

Basic reporting

Proper

Experimental design

Proper

Validity of the findings

Solid and convincing

Additional comments

It has been significantly improved and can be accepted now.

Reviewer 5 ·

Basic reporting

The authors have addressed the points of critique reasonably well.

Experimental design

The authors have addressed the points of critique reasonably well.

Validity of the findings

The authors have addressed the points of critique reasonably well.

Additional comments

The authors have addressed the points of critique reasonably well.